# A Novel Bayesian Estimation-Based Word Embedding Model For Sentiment Analysis

## Abstract

The word embedding models have achieved state-of-the-art results in a variety of natural language processing tasks. Whereas, current word embedding models mainly focus on the rich semantic meanings while are challenged by capturing the sentiment information. For this reason, we propose a novel sentiment word embedding model. In line with the working principle, the parameter estimating method is highlighted. On the task of semantic and sentiment embeddings, the parameters in the proposed model are determined by using both the maximum likelihood estimation and the Bayesian estimation. Experimental results show the proposed model significantly outperforms the baseline methods in sentiment analysis for low-frequency words and sentences. Besides, it is also effective in conventional semantic and sentiment analysis tasks.

## 1 Introduction

Word embeddings provide continuous low-dimensional vector representations of words from documents (Li et al., 2017). Aiming to capture semantic and syntactic contextual information from large datasets, the word embedding models are extensively employed to represent words in natural language processing tasks (Levy and Goldberg, 2014). For this reason, many modelling methods are proposed to generate dense representations of words (Rath, 2017).

Seeing the flourish of word embeddings, Word2vec (Mikolov et al., 2013) and GloVe (Pennington et al., 2014) are considered as the edge-cutting approaches to deal with the word contexts. C&W is another most widespread method due to the progress in neural networks (Collobert and Weston, 2008). Besides, other algorithms are integrated into the existing models. For instance, Jameel and Schockaert proposes D-GloVe by combing GloVe and Dirichlet-Multinomial language modeling (Jameel and Schockaert, 2016). More recently, the contextualized word embedding models, which improve the accuracy to a large extent, are put forward. As such, the traditional approaches are concluded as pre-trained word embeddings. Whereas, the newly proposed methods, such as ELMo by Peters M. E. et al.(Peters et al., 2018) , BERT by Devlin J. et al.(Devlin et al., 2018) and XLNet by Yang Z et al.(Yang et al., 2019) , cost large amount of computing resource for training whilst obtain a better working performance in downstream tasks. In this way, the pre-trained word embeddings still hold a great promise in handling complicated natural language processing tasks.

The aforementioned models are effective in dealing with semantic-oriented tasks. Likewise, in sentiment analysis, research is still ongoing to capture sufficient sentiment information while the sentiment embeddings typically depend on the sentiment polarity labels provided by labeled corpora to guide learning processes via objective functions (Yu et al., 2017). Tang et al. propose a method for learning sentiment embeddings by regulating the C&W model, which encodes sentiment information in the continuous word representations (Tang et al., 2015). By exploiting the prior knowledge, Li et al. incorporate the sentiment information to analyze the sentiment label of each word in target and contexts (Li et al., 2017). Maas et al. apply a semi-supervised method to get sentiment information and carry out the maximum likelihood estimation for parameter determination (Maas et al., 2011).

Notwithstanding, the pre-trained word embeddings still have challenges in tackling sentiment analysis tasks, which are concluded as the following two aspects. On the one hand, the semantically similar word may have opposite sentiment polarities. Thus the sentiment polarity identification process has to be dedicatedly designed (Tang et al., 2015) (Li et al., 2017) (Shi et al., 2018). On the other hand, the capturing of sentiment information from low-frequency words is most pronounced. Typically, the low-frequency words can be regarded as the derivation of entity nouns, new terms and some deformation high-frequency words, which also contain significant semantic information. Nevertheless, due to the low frequency, current models are absent of processing their sentiment.

The objective of this work is to devise a sentiment word embedding model. Specifically, the issue of parameter setting is deeply studied. Methods for effectively estimating the involving parameters based on Bayesian estimation and maximum likelihood estimation are proposed. For low-frequency word analysis, the Bayesian estimation is applied to determine the co-occurrence probabilities and the sentiment probabilities.

This work describes current parameter estimation approaches and the model of GloVe in Section 2, illustrates our sentiment word embedding model in Section 3, shows the experiments in Section 4, and presents the research findings in Section 5.

## 2 Preliminary

This section introduces the basic theory related to parameter estimating algorithms and the GloVe model, so as to facilitate the description of subsequent model architecture.

### 2.1 Parameter Estimating Algorithms

Typically, word vectors are taken as learning variables in the word embeddings, which results in the use of parameter estimating algorithms. The way of establishing objective function is therefore be employed. According to (Tang et al., 2014), the objective function of the Skip-Gram model is to maximize the average log probability, which is expressed as:

$$J = \frac{1}{T} \sum_{i=1}^{T} \sum_{-c \leqslant j \leqslant c, j \neq 0} \ln p\left(w_{i+j} | e_i\right) \tag{1}$$

where $T$ is the number of words in the corpus and $c$ indicates the size of window. We take $e_i$ as the embedding of target word $w_i$ and $w_{i+j}$ as the context of $w_i$. The outcome $p\left(w_{i+j}|e_i\right)$ is obtained via the hierarchical softmax. Similarly, the objective of GloVe refers to the maximum of likelihood probability and is defined as (Jameel et al., 2019):

$$J = \prod_{i,j} N\left(\ln x_{ij}; w_i \cdot \widetilde{w}_j + b_i + \widetilde{b}_j, \sigma^2\right) \tag{2}$$

where $N\left(.; \mu, \sigma^2\right)$ represents the normal distribution with the mean $\mu$ and the variance $\sigma^2$. In GloVe, the variance is determined by each word couple $(i, j)$.

In addition to objective function constructing, the estimation algorithms are applied to compute other parameters within word embeddings. In (Maas et al., 2011), the maximum posterior probability estimation identifies the parameter to weigh the semantic information (Maas et al., 2011). In D-GloVe, Jameel and Schockaert also use the semantic information weighing parameter, whose value corresponds to the Bayesian estimating outcome (Jameel and Schockaert, 2016).

### 2.2 The GloVe model

Basically, the GloVe model is a word-embedding method that combines evidence from the local context and the global counts. Typically, three distinguished words are used in this

model, which are $W_i$, $W_j$ and $W_k$. Both $W_i$ and $W_j$ are target words while $W_k$ is the context. Let $x$ be the matrix representing the word-word co-occurrence counts. We define the element $x_{ik}$ as the times for word $W_k$ appearing in the context of $W_i$. Correspondingly, $x_i = \sum_k x_{ik}$ indicates the frequency of each word occurs in the context of $W_i$. The co-occurrence probability of $W_k$ being the context word of $W_i$ is given as

$$P_{ik} = P(W_k|W_i) = x_{ik}/x_i \tag{3}$$

The parameter $P_{ik}/P_{jk}$ is taken to determine the relation of $W_i$ to $W_k$ and $W_j$ to $W_k$. For $W_k$ has a similar relation to $W_i$ and $W_j$, i.e. both relevant or irrelevant, the ratio approaches 1. The information in the ratio of co-occurrence probabilities is:

$$F\left(w_i^T \widetilde{w}_k - w_j^T \widetilde{w}_k\right) = P_{ik}/P_{jk} \tag{4}$$

where $w \epsilon \mathbb{R}^n$ refers to the target word vector and $\widetilde{w} \epsilon \mathbb{R}^n$ to the context vector. Commonly, GloVe extracts the semantic relation between different words by using the ratio of co-occurrence probabilities while the semantic information are identified via the maximum likelihood estimation (Maas et al., 2011).

## 3 Methodology

This section depicts the architecture of the sentiment word embedding, working principle of the parameter estimating process using two different estimation algorithms.

### 3.1 Sentiment Word Embedding Model Architecture

In sentiment analysis tasks, the sentiment information is captured during processing. Aiming to identify the sentiment polarities of different words, a word embedding model, incorporating the sentiment information, is established. Typically, we tend to characterize the proposed model by the loss function.

To compute the sentiment embeddings, we define the probability of $W_i$ being positive as $B_i$ and negative as $1-B_i$. Assuming that $W_i = good$ and $W_j = bad$, the value of $B_i/B_j$ is larger than 1, which indicates *good* is more positive than *bad*. In turn, the value $(1 - B_i)/(1 - B_j)$ is less than 1 since *bad* shows a negative polarity. In this way, the relations of the word sentiment are expressed by the ratio of sentiment probabilities. For $B_i + (1 - B_i) = 1$, $B_i/B_j$ and $(1 - B_i)/(1 - B_j)$ make the same sense in conveying the sentiment, we take $B_i/B_j$ to construct the sentiment relation of $W_i$ and $W_j$. More details of the words' relation and the ratios are presented in Appendix 1.

Considering the bias vector corresponds to positive sentiment polarity, we take $s \epsilon \mathbb{R}^n$ to indicate the bias vector to match the size of word vector. By transforming $W_i$ and $W_j$ into word vectors $w_i$ and $w_j$, the difference established upon $s_i$ and $s_j$ is written as:

$$F(w_i^T s_i - w_j^T s_j) = B_i/B_j \tag{5}$$

Assuming that $F$ is confirming to the homomorphisms between groups $(\mathbb{R}, +)$ and $(\mathbb{R}_{>0}, \times)$, the semantic and sentiment information is combined to get:

$$\begin{aligned}
&F\left(\left(w_i^T \widetilde{w}_k - w_j^T \widetilde{w}_k\right) + \left(w_i^T s_i - w_j^T s_j\right)\right) \\
=&F\left(w_i^T \widetilde{w}_k - w_j^T \widetilde{w}_k\right) \cdot F\left(w_i^T s_i - w_j^T s_j\right) \\
=&\frac{P_{ik}}{P_{jk}} \cdot \frac{B_i}{B_j}
\end{aligned} \tag{6}$$

Due to properties of group homomorphism, eqn.6 is transformed into

$$
\begin{aligned}
&F\left(w_i^T \widetilde{w}_k - w_j^T \widetilde{w}_k + w_i^T s_i - w_j^T s_j\right) \\
=&F\left(\left(w_i^T \widetilde{w}_k + w_i^T s_i\right) - \left(w_j^T \widetilde{w}_k + w_j^T s_j\right)\right) \\
=&\frac{F\left(w_i^T \widetilde{w}_k + w_i^T s_i\right)}{F\left(w_j^T \widetilde{w}_k + w_j^T s_j\right)} = \frac{P_{ik}}{P_{jk}} \cdot \frac{B_i}{B_j}
\end{aligned}
\tag{7}
$$

in line with

$$
F\left(w_i^T \widetilde{w}_k + w_i^T s_i\right) = P_{ik} \cdot B_i
\tag{8}
$$

According to eqn.7,the basic objective function $F$ is in the form of exponential, that is $F(x) = exp(x)$. Thus, we apply the logarithm operation to each side and have:

$$
w_i^T \widetilde{w}_k + w_i^T s_i = \ln\left(P_{ik} \cdot B_i\right) = \ln P_{ik} + \ln B_i
\tag{9}
$$

By incorporating the sentiment information, the loss function of the word embedding model is defined as

$$
loss\left(w_i, \widetilde{w}_k, s_i\right) = \sum_{i,k=1}^{V} \left[w_i^T \widetilde{w}_k + w_i^T s_i - \ln P_{ik} - \ln B_i\right]^2
\tag{10}
$$

where $V$ indicates the size of the vocabulary. The parameters $w_i^T$, $\widetilde{w}_k$ and $s_i$ are computed via gradient descent algorithms.

## 3.2 Incorporating Sentiment Information

As pointed out in the Introduction, current models use the maximum likelihood estimating algorithm for parameter determination. In this part, we preliminarily carry out the parameter estimation based on the maximum likelihood principle.

For each target word $W_i$, $x_i$ times Bernoulli experiments are conducted to extract the context independently with $V$ different outcomes in each experiment (Djuric and Huang, 2000). The occurrence number of the $k^{th}$ outcome and its probability are represented by $x_{ik}$ and $P_{ik}$. If the random variable $X_i = (X_{i1}, X_{i2}, \cdots, X_{iV})$ stands for the occurrence times of all the possibilities, i.e. $X_{ik}$ is the number for the $k^{th}$ one, the parameter $X_i$ obeys the Multinomial distribution, i.e. $X_i \sim Multinomial\left(\overrightarrow{x_i}, \overrightarrow{P_i}\right)$ with $\overrightarrow{P_i} = (P_{i1}, P_{i2}, \cdots, P_{iV})$ and $\overrightarrow{x_i} = (x_{i1}, x_{i2}, \cdots, x_{iV})$. Hence, a log-likelihood function is constructed:

$$
\begin{cases}
\max\limits_{P_{i1},P_{i2},\cdots,P_{ik},\cdots,P_{iV}} \ln L(P_{i1}, P_{i2}, \cdots, P_{ik}, \cdots, P_{iV}) \\
= \max\limits_{P_{i1},P_{i2},\cdots,P_{ik},\cdots,P_{iV}} \ln\left[(P_{i1})^{x_{i1}} \cdot (P_{i2})^{x_{i2}} \cdots (P_{ik})^{x_{ik}} \cdots (P_{iV})^{x_{iV}}\right] \\
= \max\limits_{P_{i1},P_{i2},\cdots,P_{ik},\cdots,P_{iV}} \sum\limits_{k=1}^{V} x_{ik} \cdot \ln P_{ik} \\
s.t. \sum\limits_{k=1}^{V} P_{ik} = 1
\end{cases}
\tag{11}
$$

According to eqn.11, the objective function can be resolved as an optimal problem that equality constraints. Thus, the corresponding Lagrangian function is formulated as

$$J\left(P_{i1}, P_{i2}, \cdots, P_{iV}, \lambda\right) = \sum_{k=1}^{V} x_{ik} \cdot \log P_{ik} + \lambda \left(1 - \sum_{k=1}^{V} P_{ik}\right) \tag{12}$$

where we have $P_{ik} = \frac{x_{ik}}{\lambda}$ determined by $\frac{\partial J(P_{i1}, P_{i2}, \cdots, P_{iV}, \lambda)}{\partial P_{ik}} = \frac{x_{ik}}{P_{ik}} - \lambda = 0$. Likewise, $\lambda = \sum_{k=1}^{V} x_{ik} = x_i$ is calculated with respect to $\sum_{k=1}^{V} P_{ik} = \sum_{k=1}^{V} \left(\frac{x_{ik}}{\lambda}\right) = \sum_{k=1}^{V} x_{ik}/\lambda = 1$. Thus, the estimation of $P_{ik}$ is written as

$$\widehat{P}_{ik} = x_{ik}/x_i \tag{13}$$

Notably, the obtained $P_{ik}$ is the same with that from GloVe according to eqn.3, which demonstrates the feasibility for parameter estimation in our model. In this way, the outcome of parameter sentiment probability can also be computed by using the maximum likelihood estimator. As such, a maximum likelihood estimation-based sentiment word embedding, namely MLESWE, is put forward. In this case, the Bernoulli experiments are applied to pick up the sentiment polarity of the target word $W_i$ and the outcome can be either positive or negative.

Since $B_i$ is the probability of $W_i$ being positive, we designate the distribution of $W_i$ obeys $\overrightarrow{t_i} = (t_{i0}, t_{i1})$ where $t_{i0}$ is the number of negative texts and $t_{i1}$ indicates that of the positive ones. Thus, the total number of texts including $W_i$ is expressed as $t_i = t_{i0} + t_{i1}$. Support a random variable $T_i = (T_{i0}, T_{i1})$ denotes the times of all the possibilities of outcomes and $T_i$ conforms to the binomial distribution, i.e. $T_i \sim Binomial\left(\overrightarrow{t_i}, \overrightarrow{B_i}\right)$ where $\overrightarrow{B_i} = (B_i, 1 - B_i)$. The log-likelihood function of sentiment probabilities is delivered as:

$$\max_{B_i} \ln L\left(B_i\right) = \max_{B_i} \ln \left[(B_i)^{t_{i1}} \cdot (1 - B_i)^{t_{i0}}\right] = \max_{B_i} \left[t_{i1} \cdot \ln B_i + t_{i0} \cdot \ln\left(1 - B_i\right)\right] \tag{14}$$

Similarly, $\widehat{B}_i = \frac{t_{i1}}{t_i}$ is obtained based on $\frac{\partial(\ln L)}{\partial B_i} = \frac{t_{i1}}{B_i} - \frac{t_{i0}}{1 - B_i} = 0$. Combining the semantic and sentiment information, the final loss function based on maximum likelihood principle is

$$loss\left(w_i, \widetilde{w}_k, s_i\right) = \sum_{i,k=1}^{V} \left[w_i^T \widetilde{w}_k + w_i^T s_i - \ln \frac{x_{ik}}{x_i} - \ln \frac{t_{i1}}{t_i}\right]^2 \tag{15}$$

### 3.3 Parameter Estimating using Bayesian Estimation

The Bayesian estimating method is highlighted due to its not sensitive to initialization via proper prior distributions to parameters (Ma et al., 2018). By using the prior knowledge, the deficiency of lacking information of small datasets can be resolved, which leads to the converge to the actual value (Phoong and Ismail, 2015). Accordingly, the generalization ability of the model can be improved (Wu et al., 2018). The Bayesian approach, in this way, is able to present an elegant solution for automatically determining the parameters (Ferguson, 1973). We thus employ the Bayesian estimation for the parameter estimating of the proposed model. The Bayesian estimation-based sentiment word embedding, namely BESWE, is performed.

In accordance to the assumption of maximum likelihood principle mentioned before, the prior distribution $P\left(\overrightarrow{P_i}\right)$ is assumed to obey the Dirichlet distribution of $\overrightarrow{\alpha} = (\alpha_i, \alpha_i, \cdots, \alpha_V)$.

The prior distribution is converted to $P\left(\overrightarrow{P_i}\right) = Dir\left(\overrightarrow{\alpha}\right) = \frac{\Gamma\left(\sum_k \alpha_k\right)}{\prod_k \Gamma(\alpha_k)} \prod_k P_{ik}^{\alpha_k - 1}$, with the identical likelihood function:

$$P\left(\vec{x}_i | \vec{P}_i\right) = Mult\left(\vec{x}_i, \vec{P}_i\right) = \frac{x_i!}{\prod_k^V (x_{ik}!)} \cdot \prod_k^V P_{ik}^{x_{ik}} \tag{16}$$

Considering the Dirichlet-Multinomial conjugate structure, the posterior distribution is

$$P\left(\vec{P}_i | \vec{x}_i\right) = Dir\left(\vec{\alpha} + \vec{x}_i\right) = \frac{\Gamma\left(\sum_k \alpha_k + x_{ik}\right)}{\prod_k \Gamma\left(\alpha_k + x_{ik}\right)} \cdot \prod_k P_{ik}^{\alpha_k + x_{ik} - 1} \tag{17}$$

where $\alpha_k = \lambda_1 \cdot \frac{n_k}{\sum_k n_k}$ , $n_k$ is the total number of occurrences of word $W_k$ in the corpus and $\lambda_1 > 0$ is determined by tuning data. By satisfying

$$c_{ik} = E_{P\left(\vec{P}_i | \vec{x}_i\right)} [\ln P_{ik}] \tag{18}$$

we compute the Bayesian estimating outcome of $\ln P_{ik}$ in the loss function provided by eqn.10, which is also the mean value of posterior probability in line with $P_{ik}$. As stated in (Jameel and Schockaert, 2016), the computation of $E_{P\left(\vec{P}_i | \vec{x}_i\right)} [\ln P_{ik}]$ is facilitated via Taylor expansion:

$$E_{P\left(\vec{P}_i | \vec{x}_i\right)} [\ln P_{ik}] \approx \ln E_{P\left(\vec{P}_i | \vec{x}_i\right)} [P_{ik}] - \frac{Var_{P\left(\vec{P}_i | \vec{x}_i\right)} [P_{ik}]}{2 \cdot E_{P\left(\vec{P}_i | \vec{x}_i\right)}^2 [P_{ik}]} \tag{19}$$

where we have $Var_{P\left(\vec{P}_i | \vec{x}_i\right)} [P_{ik}] = \frac{\alpha_k + x_{ik}}{\sum_k (\alpha_k + x_{ik})} \cdot \left(1 - \frac{\alpha_k + x_{ik}}{\sum_k (\alpha_k + x_{ik})}\right) \cdot \frac{1}{\sum_k (\alpha_k + x_{ik}) + 1}$ and $E_{P\left(\vec{P}_i | \vec{x}_i\right)} [P_{ik}] = \frac{\alpha_k + x_{ik}}{\sum_k (\alpha_k + x_{ik})}$. Note that $\ln P_{ik}$ is estimated via Bayesian principle in eqn.18 whose form is unlike that of eqn.13. Comparing to the maximum likelihood estimation, a direct outcome is obtained without using Laplace smoothing in experiment. Comparatively, $P\left(\vec{B}_i\right)$ is designed to obey Beta distribution with the parameter $\vec{\beta} = (\beta_0, \beta_1)$, along with the prior distribution given as

$$P\left(\vec{B}_i\right) = Beta\left(\vec{\beta}\right) = \frac{\Gamma(\beta_0 + \beta_1)}{\Gamma(\beta_0)\Gamma(\beta_1)} (1 - B_i)^{\beta_0 - 1} \cdot B_i^{\beta_1 - 1} \tag{20}$$

from which the log-likelihood function is

$$P\left(\vec{t}_i | \vec{B}_i\right) = b\left(\vec{t}_i, \vec{B}_i\right) = C_{t_i}^{t_{i1}} \cdot (1 - B_i)^{t_{i0}} \cdot B_i^{t_{i1}} \tag{21}$$

and the posterior distribution subject to the conjugate structure of Beta-Binomial is

$$P\left(\vec{B}_i | \vec{t}_i\right) = Beta\left(\vec{\beta} + \vec{t}_i\right) = \frac{\Gamma(\beta_0 + t_{i0} + \beta_1 + t_{i1})}{\Gamma(\beta_0 + t_{i0})\Gamma(\beta_1 + t_{i1})} \cdot (1 - B_i)^{\beta_0 + t_{i0} - 1} \cdot B_i^{\beta_1 + t_{i1} - 1} \tag{22}$$

where $m_k$ stands for the texts of the sentiment label $k$, $\lambda_2 > 0$ is a parameter depending on tuning data and $\beta_k = \lambda_2 \cdot \frac{m_k}{\sum_k m_k}$. Thereupon, to determine the $\ln B_i$ in eqn.10, we take the Bayesian estimation approach. The solution to the posterior probability expectation of $\ln B_i$, which is involved with $B_i$ is characterized as

$$e_i = E_{P\left(\vec{B}_i \mid \vec{t}_i\right)} \left[\ln B_i\right] \tag{23}$$

Furthermore, the Taylor expansion is employed to update the equation:

$$E_{P\left(\vec{B}_i \mid \vec{t}_i\right)} \left[\ln B_i\right] \approx \ln E_{P\left(\vec{B}_i \mid \vec{t}_i\right)} \left[B_i\right] - \frac{Var_{P\left(\vec{B}_i \mid \vec{t}_i\right)} \left[B_i\right]}{2 \cdot E^2_{P\left(\vec{B}_i \mid \vec{t}_i\right)} \left[B_i\right]} \tag{24}$$

where we have $Var_{P\left(\vec{B}_i \mid \vec{t}_i\right)} \left[B_i\right] = \frac{\beta_1 + t_{i1}}{\sum_k (\beta_k + t_{ik})} \cdot \left(1 - \frac{\beta_1 + t_{i1}}{\sum_k (\beta_k + t_{ik})}\right) \cdot \frac{1}{\sum_k (\beta_k + t_{ik}) + 1}$ and $E_{P\left(\vec{B}_i \mid \vec{t}_i\right)} \left[B_i\right] = \frac{\beta_1 + t_{i1}}{\sum_k (\beta_k + t_{ik})}$. Hence, the final loss function of BESWE can be obtained:

$$loss\left(w_i, \widetilde{w}_k, s_i\right) = \sum_{i,k=1}^{V} \left[w_i^T \widetilde{w}_k + w_i^T s_i - c_{ik} - e_i\right]^2 \tag{25}$$

## 4 EXPERIMENTS

In this section, the working performance of BESWE and MLESWE are evaluated. The task of word similarity analysis is carried out. To deliver the sentiment embeddings, both word- and sentence-level sentiment analysis using different models is conducted.

### 4.1 EXPERIMENT SETTINGS

**Datasets.** The dataset SST (Stanford Sentiment Tree) is employed for the mode training. There are five classes annotations within SST, which are very negative, negative, neutral, positive and very positive. Typically, we assign the value 3 and 4 to represent the positive polarity, 0 and 1 to negative and 2 to else. For our models, the word representation dimension is 50, the learning rate is 0.05 and the iteration number is 50. Besides, the loss function is optimized with the deployment of AdaGrad.

**Baseline Methods.** We evaluate the proposed model in comparison to other state-of-the-art models. The models of word embeddings, such as C&W, word2vec and GloVe, together with models of sentiment embeddings, including SE-HyRank and DLJT2, are implemented. For the baseline methods, we use default settings in the provided implementations or described as their papers and the word representation dimension is 50.

**Word Similarity.** Computing word similarity (Levy et al., 2015) aims to capture the general meanings of words. In this research, the word similarity tasks are conducted on the dataset EN-WS-353-ALL, EN-WS-353-SIM and SCWS, which are detailed illustrated in (Jameel et al., 2019).

**Word-level Sentiment Analysis.** We conduct word-level sentiment analysis on two sentiment lexicons, namely MPQA and NRC. The number of positive and negative items for MPQA is 2301 and 4151 while for NRC is 2231 and 3324. The N-fold cross validation with N=5 and N=10 is performed. An SVM classifier is trained whose average accuracy is the evaluation metric. Specifically, the words from SST corpus are extracted and converted into word embeddings, which are taken as the features of SVM.

As Bayesian estimating principle is capable of tackling low-frequency words, we distinctively pick up the words with a frequency less than 5 for analysis. Statistically, the SST corpus contains 9984 low-frequency words.

**Sentence-level Sentiment Analysis.** Considering the sentiment analysis for sentence, the movie review polarity datasets MovieReview is employed (Pang and Lee, 2005), which contains 10622 samples with the proportion of each polarity 1:1. We use a convolutional neural network (CNN) model, namely Text-CNN, with its online implementation (Kim, 2014). Likewise, the inputs of Text-CNN are word embeddings as well. The training episode is set as 200 epochs using the default settings.

Similarly, we also pick the low-frequency words with the occupation over 10% as the low-frequency sentences for testing. There are totally 1258 sentences cater to the demands and are all sent to Text-CNN classifier for processing.

## 4.2 Experimental Results

**Word Similarity.** On the task of working performance evaluation, we first present the results for of word similarity analysis (Fig. 1). It can be observed BESWE outperforms other algorithms on all datasets, indicating that our model is capable to capture sufficient semantic information. Distinctively, the implementation of MLESWE, although not as good as BESWE, still achieves a better result on the average accuracy (i.e. Ave_Acc in Fig. 1) than the original GloVe. Yet the maximum likelihood estimating algorithm can also be applied to parameter determination of the word embeddings.

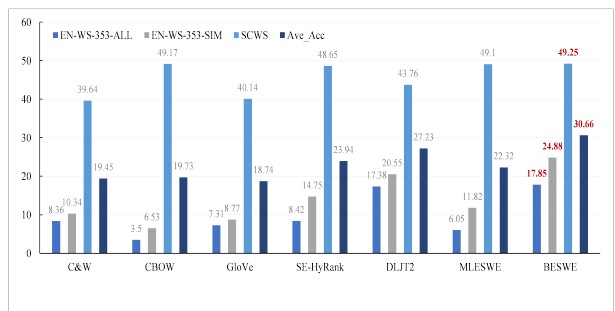

Figure 1: Word similarity results in Spearman rank correlation

**Word-level Sentiment Analysis Results.** The word-level sentiment analysis task is conducted on the dataset of single-word entries. The DLJT2 model outperforms other word embedding models by incorporating sentiment information into the learning processes, as shown in Fig.2. Compared to the state-of-the-arts, our model fails to exceed the outcome of the best method on average accuracy. Encouragingly, the BESWE model shows an even better performance in tackling the low-frequency words.

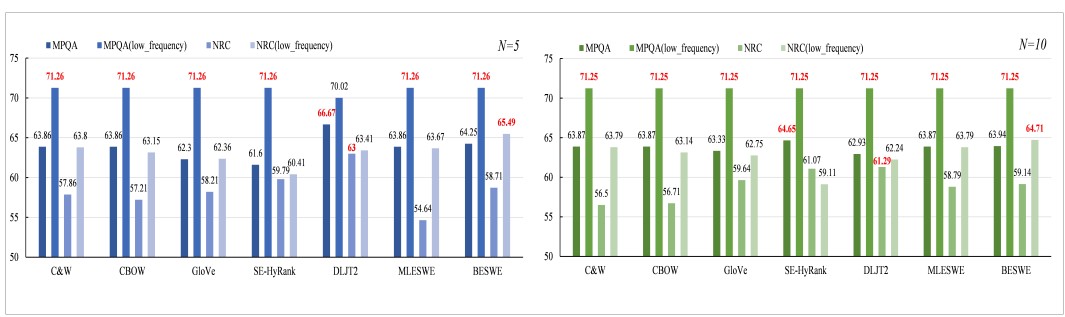

Figure 2: Word-level sentiment analysis results

**Sentence-level Sentiment Analysis Results.** The working performance of the proposed model is further evaluated on the sentence-level sentiment analysis task. From Fig.3, we see that SE-HyRank has a better performance than any other algorithms in average accuracy. Clearly, the outcome of BESWE is anyhow decent which is comparable with that of SE-HyRank. Regarding low-frequency sentences, the minimum performance gap of over 9% against SE-HyRank is reported. Consequently, for the sentiment analysis of low-frequency words or low-frequency sentences, BESWE always obtain the best and most consistent results in the identification of sentiment polarity.

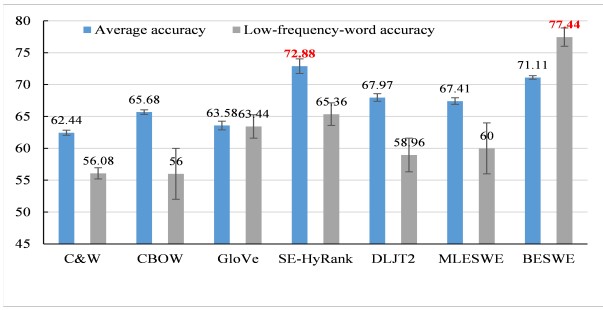

Figure 3: Sentence-level sentiment analysis results

*Effects of $\lambda_1$ and $\lambda_2$.* The hyperparameters in BESWE, i.e. regulatory factors $\lambda_1$ and $\lambda_2$, are used to represent the semantic and the sentiment information. The settings of the involving parameter can be therefore determined. The values of $\lambda_1$ and $\lambda_2$ are varied within the collection of $\{1, 0.75, 0.5, 0.25, 0.1, 0.05, 0.02, 0.01\}$. Firstly, the value of $\lambda_1$ is set as $\{1, 0.75, 0.5, 0.25, 0.1, 0.05, 0.02, 0.01\}$. When $\lambda_2 = 1$, we name BESWE as BESWE#1 and $\lambda_2 = 0.75$ as BESWE#2, and so on so forth. Correspondingly, the value of $\lambda_2$ is also picked from the same set and named from BESWE#9 to BESWE#16 in the same order. Totally, we get 16 different models.

The results on the sentence-level sentiment analysis against different hyperparameter settings are shown in Fig.4(a) and Fig.4(c). Likewise, the results for low-frequency sentences are in Fig.4(b) and Fig.4(d). We take LowFreSentence#n to nominate the outcomes from low-frequency sentences.

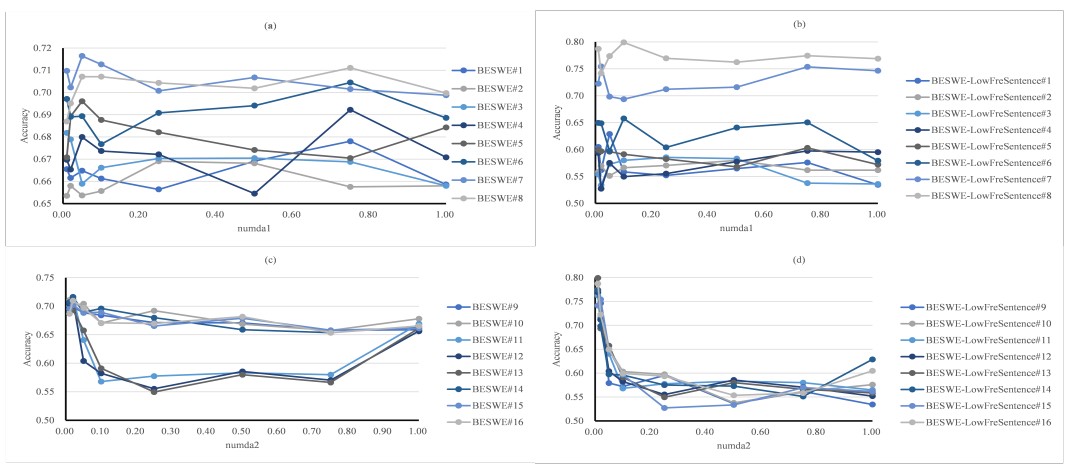

Figure 4: Sensitivity of $\lambda_1$ and $\lambda_2$ on BESWE with Sentence-level Sentiment Analysis

The sentence-level sentiment analysis reaches the highest accuracy 71.64% at the point $\lambda_1 = 0.05$ and $\lambda_2 = 0.02$. For the analysis of low-frequency sentences, the optimal values of $\lambda_1$ and $\lambda_2$ are 0.1 and 0.01, which results in an accuracy of 79.92%.

The experimental results verify the effectiveness of the proposed sentiment word embedding. The BESWE model outperforms other state-of-the-art in word similarity identification. In the sentiment analysis of both word level and sentence level, our method still presents comparable outcomes. Specifically, by integrating the prior knowledge into sentiment probabilities estimating, the BESWE model is a better alternative for low-frequency-word sentiment capturing. It is reasonable to expect better performance in sentiment analysis tasks, as it is the case.

## 5 Conclusion

In this work, the designing and deploying of the sentiment word embeddings is deeply studied. On the foundation of current word embedding models, the estimation principle of the objective function, together with other parameters, are investigated. Motivated by the significance of sentiment information, a novel word embedding model for sentiment analysis is established.

Within the proposed model, both semantic and sentiment information is integrated into the word vectors. Aiming to construct the objective function, the group homomorphism theory is applied. As for the parameter determination, the maximum likelihood estimator and the Bayesian estimator are employed. Experiments are conducted on various tasks to evaluate the working performance. In comparison to the baseline models, our model is capable of tackling word similarity tasks. For the purpose of sentiment embeddings representation, the proposed model is effective in word-level and sentence-level sentiment analysis. Specifically, it outperforms other methods on low-frequency words and sentences sentiment polarity identification to demonstrate its efficacy.

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

# A  APPENDIX I

For $W_i = good$ and $W_j = bad$,

we have $B_i/B_j > 1$,

$(1 - B_i)/(1 - B_j) < 1$,

i.e. $W_i = good$ is more positive than $W_j = bad$.

For $W_i = good$ and $W_j = great$,

we have $B_i/B_j \approx 1$,

$(1 - B_i)/(1 - B_j) \approx 1$,

i.e. $W_i = good$ and $W_j = great$ are of positive polarities.

For $W_i = then$ and $W_j = home$,

we have $B_i/B_j \approx 1$,

$(1 - B_i)/(1 - B_j) \approx 1$,

i.e. $W_i = then$ and $W_j = home$ are of neutral polarities.

The specific sentiment probabilities calculated by maximum likelihood estimation are presented in Table 1.

Table 1: Sentiment probabilities of different words using maximum likelihood estimation

| Probability and Ratio | $W_i = "good"$ $W_j = "bad"$ | $W_i = "good"$ $W_j = "great"$ | $W_i = "then"$ $W_j = "home"$ |
|---|---|---|---|
| $B_i$ | 0.5879 | 0.5879 | 0.3855 |
| $B_j$ | 0.0905 | 0.7063 | 0.4909 |
| $B_i/B_j$ | 6.4944 | 0.8323 | 0.7854 |
| $(1 - B_i)/(1 - B_j)$ | 0.1540 | 1.2015 | 1.2733 |

Likewise, the outcomes based on Bayesian estimation are in Table 2.

Table 2: Sentiment probabilities of different words using Bayesian estimation

| Probability and Ratio | $W_i = "good"$ $W_j = "bad"$ | $W_i = "good"$ $W_j = "great"$ | $W_i = "then"$ $W_j = "home"$ |
|---|---|---|---|
| $B_i$ | 0.5872 | 0.5872 | 0.3819 |
| $B_j$ | 0.0886 | 0.7053 | 0.4864 |
| $B_i/B_j$ | 6.6272 | 0.8326 | 0.7852 |
| $(1 - B_i)/(1 - B_j)$ | 0.1509 | 1.2011 | 1.2735 |

