# OpenReview forum: "A novel Bayesian estimation-based word embedding model for sentiment analysis"
_ICLR.cc/2020/Conference — Reject_

### Official Review · AnonReviewer1 · 2019-10-23
**Official Blind Review #1**

**Rating:** 6

**Review:**

The paper aims at extending GloVe word embedding model so that the resulting embeddings should capture sentiments (e.g. "good" is positive while "bad" is negative). The key idea is to employ an extension term to deal with the fact that some words appearing in text with sentiment information. Furthermore, to deal with the fact that many words an infrequent, besides maximum likelihood estimation, the paper proposes to use bayesian estimation. In the experiments, Stanford sentiment tree (SST) corpus is used. The word embeddings from the two models (each trained on different estimation methods) show their capability of expressing sentiments, compared with popular methods like Glove, word2vec.

I would accept this paper because:
- This paper is well written, with thoughtful maths details.
- The proposed models, although are extensions of GloVe, gives interesting (and rigorous) points of how to add sentiment information.
- The experiments do support what the paper claims.

I would reject it because of the experiments. The dataset (SST) is so small and thus is questionable about the quality of the learned word embeddings and the comparisons. I think there should be better ways, such as word embeddings are trained on massive data (like for GloVe and word2vec), then are fine-tuned on sentiment analysis dataset. Also, I was wondering whether there's a way to collect more sentiment data (like in SE-HyRank paper).



**Experience Assessment:**

I do not know much about this area.

**Review Assessment: Checking Correctness Of Derivations And Theory:**

I assessed the sensibility of the derivations and theory.

**Review Assessment: Checking Correctness Of Experiments:**

I assessed the sensibility of the experiments.

**Review Assessment: Thoroughness In Paper Reading:**

I read the paper at least twice and used my best judgement in assessing the paper.

---

> ### Author Response · Authors · 2019-11-13
> **Initial Response to Review #1**
>
> Thank you for your insightful comments.
>
> Our model is currently applied to the corpus of labeled samples. The widely-used models, such as word2vec and GloVe, are trained by the unlabeled samples, which are infinite for using. Since the datasets with labeled samples are of high cost to make, the commonly used labeled dataset are SST，IMDb，Yelp P. and Yelp F. at this stage. By comparing the datasets and referring to previous work [1], we decide to use SST for model training.
>
> Thank you for your suggestion. We will consider more training corpus in the future research.
>
> References
> [1] Yang Li, Quan Pan, Tao Yang, Suhang Wang, Jiliang Tang, and Erik Cambria. Learning word representations for sentiment analysis. Cognitive Computation, 9(6):843–851, 2017.

---

### Official Review · AnonReviewer2 · 2019-10-23
**Official Blind Review #2**

**Rating:** 1

**Review:**


This paper proposes a method to learn word embedding by incorporating additional sentiment information. The proposed method extends from D-GloVe by adding the probability of positive sentiment to the loss function. The paper presents three experiments: word similarity, word-level sentiment analysis, and sentence-level sentiment analysis. The experiments show that the method performs comparably with other baseline methods and outperforms in the low-frequency sentence setting (i.e. sentence containing lower frequency words).

I recommend rejecting this paper because (1) the writing is unclear and hard to follow, and (2) the experiment results are not convincing.

From what I can understand in the model part, there are many clarifications needed, not to mention the writing style. I think the re-derivations of GloVe and D-GloVe are not helpful as they cloud the main contribution of the paper. The author should clearly highlight the differences between the main subjects of the experiment: MLESWE and BESWE. In addition, it is not clearly motivated why we need Dirichlet prior for the sentiment variable.

While the claim is to learn better embeddings for rare words, the experiments show that the proposed methods have similar results to the previous work. The only gain we can observe is in the sentence-level experiments in which other factors could affect the performance. Thus, it is hard to draw a supportive conclusion.

Finally, the writing quality must be improved. The paper contains a lot of unrelated and redundant texts (it could be that I could follow the paper).
1. I do not think eq 2 is a representative of how the paper train the model, nor attempt to compare with.

2. As mention earlier, in section 3.2 and 3.3, the re-derivation is not particularly helpful. I think the paper should put more emphasis on the novelty of the work.

3. Plots in the experiment results are illegible. Tables should be more suitable for Figures 1, 2, and 3.

I urged the authors to revise this paper and make sure it follows the formatting guideline, especially the citations. Finally, I'd recommend the authors have a professional writer (English) review the paper before submission.


**Experience Assessment:**

I have read many papers in this area.

**Review Assessment: Checking Correctness Of Derivations And Theory:**

I did not assess the derivations or theory.

**Review Assessment: Checking Correctness Of Experiments:**

I assessed the sensibility of the experiments.

**Review Assessment: Thoroughness In Paper Reading:**

I read the paper at least twice and used my best judgement in assessing the paper.

---

> ### Author Response · Authors · 2019-11-13
> **Initial Response to Review #2**
>
> Thank you for your insightful comments. We address your concerns in detail below.
>
> Reviewer 2: “...why we need Dirichlet prior for the sentiment variable.”
>
> Since we mainly work on the sentiment information of the low-frequency words, the Bayesian estimation is taken to compute the sentiment probabilities and incorporate them into the loss function. By using the prior information, the small samples such as low frequency words with less information, are used to better converging to the true value [1].
>
> The reason for using Dirichlet prior is that the Dirichlet distribution is a conjugate prior of multiple distributions in Bayesian probability theory. In the manuscript, we also mention the Dirichlet-Multinomial conjugate structure. In addition, the Beta prior corresponds to the Beta distribution, which is a conjugate prior to the binomial distribution.
>
> Reviewer 2: “While the claim is to learn better embeddings for rare words, the experiments show that the proposed methods have similar results to the previous work. The only gain we can observe is in the sentence-level experiments in which other factors could affect the performance. Thus, it is hard to draw a supportive conclusion.
> ”
>
> In the experiments, we have to admit that the word-level sentiment analysis for low-frequency words has a similar outcome compared to the previous work in MPQA lexicons. However, for the NRC sentiment lexicons, the BESWE outperforms other baseline models. Further, in the testing of Sentence-level sentiment for low-frequency sentences and the Word similarity, our method is proved to be a comparable alternative. Therefore, the proposed model BESWE validates its effectiveness in computing the sentiment probabilities. Yet it is impossible to have a perfect experiment to cover all the properties of the proposed model. We just focus on the aspects we are interested in at this stage. More research is still ongoing to show its working performance and in turn optimize the current model.
>
> Reviewer 2: “I do not think eq 2 is a representative of how the paper train the model, nor attempt to compare with.
> ”
>
> Eq 2 in this paper is to show how to convey the GloVe model by using the maximum likelihood estimation instead of representing the training model of our method. By integrating the parameter estimation principle and the word embedding model in previous work, the proposed method can be illustrated in a easier manner.
>
> Reviewer 2: “..., in section 3.2 and 3.3, the re-derivation is not particularly helpful. I think the paper should put more emphasis on the novelty of the work. ”
>
> The co-occurrence probability derivation in Section 3.2 tend to better express the co-occurrence probabilities based on the principle of maximum likelihood estimation. This part is valuable for delivering our model but is not illustrated in GloVe. Other equations in Section 3.2 and 3.3 are for comprehensively deriving the loss function, incorporating the sentiment information to our model. Notably, the employment of the sentiment information is a novel process in our work.
>
> Once again, thanks for your time and your attention.
>
> References
> [1] Phoong S Y, Ismail M T. A Comparison Between Bayesian and Maximum Likelihood Estimations in Estimating Finite Mixture Model for Financial Data[J]. Sains Malaysiana, 2015, 44(7): 1033-1039.

---

### Official Review · AnonReviewer3 · 2019-10-24
**Official Blind Review #3**

**Rating:** 3

**Review:**

The paper proposed a word embedding model to incorporate the sentiment information. The paper provided both maximum likelihood estimation and maximum posterior estimation for the proposed framework. Improved experiment results on word similarity and low frequency embeddings are presented. Overall, the paper incorporates the sentiment information in a neat way. And my main concern is the around the Bayesian inference and the prior knowledge distilled into the model.  Detail comments are as following,

1. The model employed Laplace approximation for posterior distribution. Not quite sure this is a good idea for the Bernoulli case since Laplace approximation is trying to use Gaussian distribution to approximate the region around the mode. How will the MAP solution compare with a full Bayesian solution such as VB or sampling-based methods?

2. Another concern is the prior introduced into the model. Normally prior information will be washed away as the training data grow. Not the case for the low frequency examples that the model performed well on. Would it possible that the improved performance on low frequency example is just a side effect of the biased introduced by the prior? How sensitive will the embedding perform with respect to the prior selected?


**Experience Assessment:**

I have read many papers in this area.

**Review Assessment: Checking Correctness Of Derivations And Theory:**

I assessed the sensibility of the derivations and theory.

**Review Assessment: Checking Correctness Of Experiments:**

I assessed the sensibility of the experiments.

**Review Assessment: Thoroughness In Paper Reading:**

I read the paper at least twice and used my best judgement in assessing the paper.

---

> ### Author Response · Authors · 2019-11-13
> **Initial Response to Review #3**
>
> Thank you for your insightful comments. We address your concerns in detail below.
>
> Reviewer 3: “The model employed Laplace approximation for posterior distribution. Not quite sure this is a good idea for the Bernoulli case since Laplace approximation is trying to use Gaussian distribution to approximate the region around the mode. ”
>
> Indeed, the Laplace approximation, VB and sampling-based methods are three most widely-used algorithms for resolving the probability density functions. Whereas, our work mainly focuses on the issue of parameter estimation, which aims to compute the value of co-occurrence probabilities and sentiment probabilities. For the approaches of parameter estimation, the probability density function does not have to obey the Gaussian distribution. Besides, for each target word, extracting its contexts and sentiment polarities can be regarded as independent random experiments.
>
> Reviewer 3: “How will the MAP solution compare with a full Bayesian solution such as VB or sampling-based methods?”
>
> In this work, the primary issue to address is the parameter estimation instead of the probability density function. As such, the VB and sampling-based methods are not considered currently.
>
> Reviewer 3: “Normally prior information will be washed away as the training data grow. Not the case for the low frequency examples that the model performed well on. ”
>
> I don't particularly understand what you mean?
>
> Reviewer 3: “Would it possible that the improved performance on low frequency example is just a side effect of the biased introduced by the prior? ”
>
> The introduction of the prior information can certainly cause the working performance improvement. In this work, the Bayesian estimating method is taken to extract the sentiment information of the low frequency words and proved to be effective. By using the prior knowledge, the deficiency of lacking information of small datasets such as low frequency words can be resolved, which leads to the converge to the actual value [1].
>
> Reviewer 3: “How sensitive will the embedding perform with respect to the prior selected? ”
>
> Little attention has been paid to the prior selected issue. Anyway, this issue will be concerned in the following studies.
>
> Once again, thanks for your time and your attention.
>
> References
> [1] Phoong S Y, Ismail M T. A Comparison Between Bayesian and Maximum Likelihood Estimations in Estimating Finite Mixture Model for Financial Data[J]. Sains Malaysiana, 2015, 44(7): 1033-1039.

---

### Decision · Program_Chairs · 2019-12-19

**Decision:**

Reject

**Comment:**

This paper proposes a method to improve word embedding by incorporating sentiment probabilities. Reviewer appreciate the interesting and simple approach and acknowledges improved results in low-frequency words.

However, reviewers find that the paper is lacking in two major aspects:
1) Writing is unclear, and thus it is difficult to understand and judge the contributions of this research.
2) Perhaps because of 1, it is not convincing that the improvements are significant and directly resulting from the modeling contributions.

I thank the authors for submitting this work to ICLR, and I hope that the reviewers' comments are helpful in improving this research for future submission.